# Decomposed Prompt Tuning via Low-Rank Reparameterization

**Yao Xiao[1]** and **Lu Xu[1]** and **Jiaxi Li[1]** and **Wei Lu[1]** and **Xiaoli Li[2]**

[1]Singapore University of Technology and Design

[2]Institute for Infocomm Research, A*Star, Singapore

{xiao_yao, xu_lu, jiaxi_li}@mymail.sutd.edu.sg

luwei@sutd.edu.sg, xlli@i2r.a-star.edu.sg

## Abstract

While prompt tuning approaches have achieved competitive performance with high efficiency, we observe that they invariably employ the same initialization process, wherein the soft prompt is either randomly initialized or derived from an existing embedding vocabulary. In contrast to these conventional methods, this study aims to investigate an alternative way to derive soft prompt. Our empirical studies show that the soft prompt typically exhibits a low "intrinsic rank" characteristic. With such observations, we propose *decomposed prompt tuning*, a novel approach that utilizes low-rank matrices to initialize the soft prompt. Through the low-rank reparameterization, our method significantly reduces the number of trainable parameters while maintaining effectiveness. Experimental results on the SuperGLUE benchmark in both high-resource and low-resource scenarios demonstrate the effectiveness of the proposed method.[1]

## 1 Introduction

Pre-trained language models (Peters et al., 2018; Radford et al., 2019; Devlin et al., 2019; Liu et al., 2020; Raffel et al., 2020) have achieved remarkable performance on various natural language understanding and generation tasks. The *pretrain-then-finetune* paradigm has been adopted as a common approach to deal with downstream tasks. However, such a paradigm is often considered inefficient, especially in the era of large language models (LLMs), as it requires tuning a large number of model parameters and saving a separate model for each task.

Recently, parameter-efficient fine-tuning (PEFT) approaches (Houlsby et al., 2019; mahabadi et al., 2021; Liu et al., 2022b,a; Vu et al., 2022; Asai et al., 2022; Wang et al., 2022, 2023) have been proposed to address this challenge. The main idea

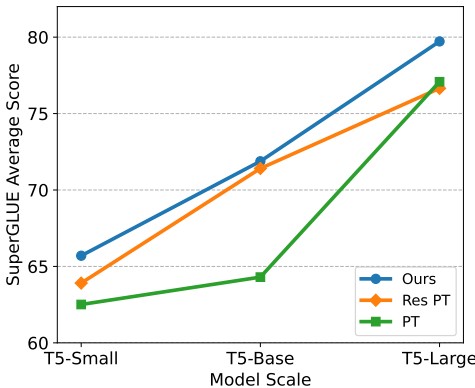

Figure 1: Average performance over all the datasets of SuperGLUE with T5 models. The number of trainable parameters used are 11.2K, 102K, and 925K with T5-Large for DPT, vanilla prompt tuning (PT) (Lester et al., 2021), and Residual PT (Res PT) (Razdaibiedina et al., 2023), respectively. More details are included in Section 5.1.

of these methods is to fine-tune only a subset of the model's parameters or additionally introduced parameters while freezing the majority of parameters of a pre-trained model. These approaches require saving only the trainable parameters for different tasks, striking a balance between performance and efficiency.

The success of PEFT methods is also aligned with the previous findings that pre-trained models possess a low "intrinsic rank" (Li et al., 2018; Aghajanyan et al., 2020). Aghajanyan et al. (2020) empirically showed that employing a low-dimensional reparameterization is equally effective as full-model fine-tuning. Hu et al. (2022) further demonstrated that weight updates during model training also exhibit a low "intrinsic rank". By only tuning the proposed low-rank adaptation module, their approach achieves high efficiency while maintaining competitive performance.

Another line of research work focuses on P*-

---

[1]Our code is available at https://github.com/XYaoooo/DPT.

tuning (Li and Liang, 2021; Liu et al., 2021; Qin and Eisner, 2021; Lester et al., 2021). Specifically, Li and Liang (2021) proposed prefix tuning by prepending a sequence of virtual tokens to each transformer layer, and it updates the representations of these virtual tokens while keeping the pre-trained model frozen. Prompt tuning (Lester et al., 2021) further simplified prefix tuning by updating only a sequence of continuous prompt tokens in the embedding layer.

Owing to its simplicity, subsequent studies (Ma et al., 2022; Razdaibiedina et al., 2023) continuously improve the vanilla prompt tuning approach. Despite the substantial achievements of these prompt tuning methods, we observe that they invariably employ the same initialization process, wherein the soft prompt is either randomly initialized or derived from the existing embedding vocabulary. Different from the previous approaches, this paper aims to explore an alternative approach to deriving soft prompt.

Our motivation stems from the observations that weight updates in the transformer layer during training have a low "intrinsic rank", as highlighted by Hu et al. (2022). This leads us to inquire whether soft prompt also have a similar low "intrinsic rank" pattern. To this end, we conduct studies examining the "intrinsic rank" of soft prompt, and we include the details of the analysis in Section 2. Based on the studies, we find that the soft prompt indeed tends to exhibit a low "intrinsic rank" behavior. Armed with this insight, we propose *decomposed prompt tuning* (DPT), a novel approach that employs low-rank matrices to initialize the soft prompt. Specifically, we decompose the original soft prompt into the product of two compact matrices, and we include the detailed description in Section 3. With this low-rank reparameterization, DPT significantly reduces the number of trainable parameters while achieving strong performance. A comparison with previous approaches, depicted in Figure 1, verifies the efficacy of our method.

Our contribution can be summarized as follows:

- We present an empirical study to show that the soft prompt exhibits a low "intrinsic rank" characteristic.

- Motivated by such findings, we propose our method to initialize the soft prompt with low-rank matrices. Experimental results on the SuperGLUE benchmark in both high-resource

and low-resource scenarios demonstrate the effectiveness of our proposed approach.

- Compared with the vanilla prompt tuning approach, our method requires significantly fewer trainable parameters while achieving strong performance (i.e., 11.2K vs. 102K with T5-Large (Raffel et al., 2020)).

## 2 Emergence of "Intrinsic Rank"

To investigate the rank of soft prompt, we design an analysis based on the vanilla prompt tuning approach on the sub-tasks within the SuperGLUE benchmark. Specifically, we re-design the soft prompt, which facilitates a probe into its rank.

### 2.1 Prompt Tuning

We employ the vanilla prompt tuning (Lester et al., 2021) approach to conduct our analysis, as illustrated on the left side of Figure 2. By considering the classification task as a conditional generation task, prompt tuning models the probability as:

$$\Pr{}_{\theta; \theta_P}(Y|[P; X]) \qquad (1)$$

where the input is the concatenation of the prepended prompt $P$ of length $c$ and the original input $X$ of length $n$. After the embedding layer, the representations for the prompt and the original input are $P_{emb} \in \mathbb{R}^{e \times c}$ and $X_{emb} \in \mathbb{R}^{e \times n}$, respectively, and $e$ is the embedding dimension of the pre-trained model. $Y$ denotes a sequence of output tokens, which also serves as the class label for the input[2]. $\theta$ indicates the frozen parameters of the pre-trained model, while $\theta_P$ represents the trainable parameters corresponding to the prompt $P$.

### 2.2 Low-Rank Behavior of Soft Prompt

As highlighted earlier, the soft prompt $P_{emb} \in \mathbb{R}^{e \times c}$ comprises the only tunable parameters during training. Typically, such a weight matrix in a neural network has a full rank both before and after model training. Therefore, to investigate whether the prompt matrix $P_{emb}$ has a low "intrinsic rank", we need to reparameterize the soft prompt.

Consider an example matrix $M \in \mathbb{R}^{m \times n}$, the singular value decomposition of the matrix $M$ is in the form of $M = U \Sigma V$, where $U \in \mathbb{R}^{m \times m}$, $V \in \mathbb{R}^{n \times n}$, and $\Sigma \in \mathbb{R}^{m \times n}$ is a diagonal matrix. Note that the diagonal entries of $\Sigma$ represent the singular values of $M$, and the number of non-zero

---

[2]$Y$ can be a single token or a sequence of tokens.

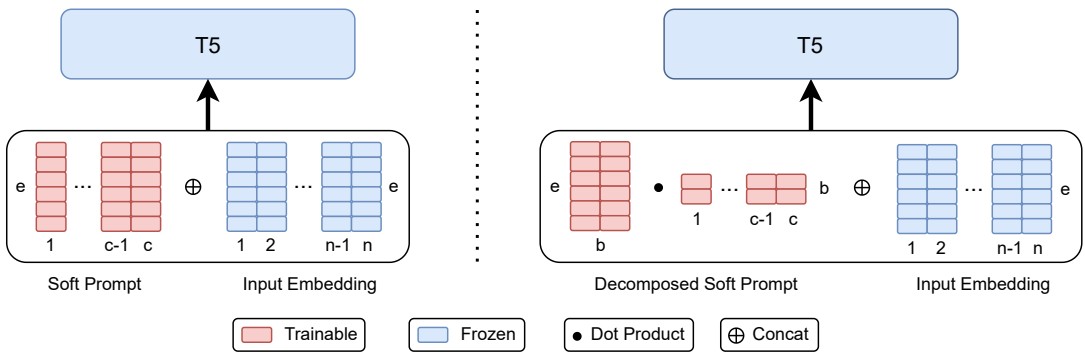

Figure 2: Left: vanilla prompt tuning, Right: our proposed DPT. The soft prompt $P_{emb} \in \mathbb{R}^{e \times c}$ (left) can be decomposed into two matrices, $A \in \mathbb{R}^{e \times b}$ and $B \in \mathbb{R}^{b \times c}$. By setting $b \ll min(c, e)$, the number of trainable parameters of our DPT ($eb + bc$) is much smaller than that of vanilla prompt tuning ($ec$).

singular values is equal to the rank of $M$. With the above decomposition, we can first reparameterize the soft prompt $P_{emb} \in \mathbb{R}^{e \times c}$ as:

$$P_{emb} = U\Sigma V \qquad (2)$$

Here, the matrices $U \in \mathbb{R}^{e \times e}$ and $V \in \mathbb{R}^{c \times c}$ are randomly initialized and set as trainable parameters in this analysis. $\Sigma \in \mathbb{R}^{e \times c}$ is initialized with positive values along the diagonal, while the off-diagonal elements are set as 0. We further impose constraints on $\Sigma$ such that only the diagonal entries are trainable while the remaining are frozen. Intuitively, if the soft prompt $P_{emb}$ has a low "intrinsic rank", then some of the diagonal entries of $\Sigma$ will converge to 0 during training. We provide an explanation of why we can use the rank of $\Sigma$ as an approximation for the rank of $P_{emb}$ in Appendix A.1.

However, due to the dense nature of the neural network, the diagonal entries of $\Sigma$ can hardly be updated to exact 0. To resolve this challenge, we apply a rectified linear unit (ReLU) to the diagonal matrix $\Sigma$:

$$P_{emb} = U\texttt{ReLU}(\Sigma)V \qquad (3)$$

Through such a reparameterization of the soft prompt $P_{emb}$, we can count the number of positive entries in the rectified diagonal matrix $\Sigma$ post-training to approximately investigate the rank of the matrix $P_{emb}$. We can infer that if the soft prompt has a low "intrinsic rank", more diagonal entries of $\Sigma \in \mathbb{R}^{e \times c}$ will be updated to negative values.

We conduct our empirical experiments with the T5-Base model on the CB dataset (De Marneffe et al., 2019) of SuperGLUE benchmark, and Figure 3 shows the results. We observe that the number of positive diagonal entries decreases while the

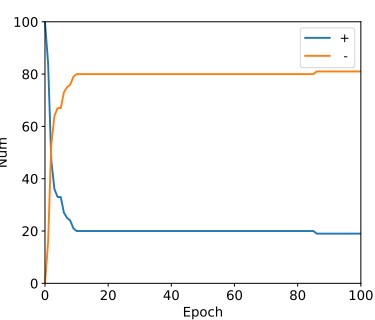

Figure 3: Number of positive and negative diagonal entries in $\Sigma$.

number of negative diagonal entries increases as training progresses. We interpret these observations as indicative that the soft prompt has a low "intrinsic rank". Note that the soft prompt length $c$ is set as the commonly used value of 100 in this analysis, this accounts for the initial count of 100 positive diagonal entries. We also observe similar behavior on other datasets with models of different sizes, with additional details provided in Appendix A.2.

## 3 Method

Motivated by the observation of the soft prompt's low "intrinsic rank" behavior, as discussed in the previous section, we propose a parameterization of the soft prompt that explicitly constrains it to be of low rank. In this section, we provide the details of our approach, as illustrated on the right side of Figure 2.

### 3.1 Decomposition of Soft Prompt

As described in Section 2.1, the input of the vanilla prompt tuning approach is the concatenation of the

prompt $P$ and the input text $X$. The overall input to the pre-trained model is then as follows:

$$\{p_1, p_2, \ldots, p_c, x_1, x_2, \ldots, x_n\} \quad (4)$$

where there are $c$ soft prompt tokens and $n$ text tokens. The focus of our approach is a more efficient representation of the soft prompt $P_{emb}$, which usually has dimension $e \times c$.

Instead of using a random initialization for the soft prompt, we decompose it into the product of two matrices:

$$P_{emb} = AB \quad (5)$$

where $A \in \mathbb{R}^{e \times b}$ and $B \in \mathbb{R}^{b \times c}$, as illustrated on the right side of Figure 2. One of the main advantages of this representation is that it offers us the ability to modulate the rank of the soft prompt $P_{emb}$ by controlling the size of the intermediary dimension $b$, which we term as the "bottleneck". Specifically, by setting $b$ to a relatively small value, the resulting soft prompt $P_{emb}$ inherently possesses a low rank. Moreover, this decomposition approach leads to a reduction in the number of trainable parameters by compressing the bottleneck. Specifically, there are $ec$ trainable parameters in the vanilla prompt tuning approach. With our proposed DPT approach, the number of trainable parameters is $eb + bc$. When setting $b \ll min(c, e)$, the total number of trainable parameters can be significantly reduced. Note that $c$ is usually set to 100 and $e$ is 1024 for T5-Large (Raffel et al., 2020), and we set $b$ as 10 in our main experiments. A more detailed analysis of bottleneck $b$ can be found in Section 6.1.

### 3.2 Training and Inference

We adopt similar training procedures as discussed in Section 2.1. Our training objective is as follows:

$$\Pr_{\theta; \theta_P}(Y|[P; X]) \quad (6)$$

where $P$ indicates the prompt with a length $c$ tokens, $\theta$ denotes the frozen parameters of the pre-trained model, and $\theta_P$ is the trainable parameter associated with the prompt $P$. In our approach, the representation of the soft prompt $P_{emb}$, is derived as the product of two matrices (Eq. 5), $A$ and $B$, which are both initialized randomly and serve as the tunable parameters of our model. $\theta_P$ corresponds to the parameters contained in matrices $A$ and $B$. After the training is completed, we can store the product of $A$ and $B$ for inference.

## 4 Experimental Setup

### 4.1 Datasets

To assess the effectiveness of our proposed DPT method, we conduct experiments on eight datasets from the SuperGLUE benchmark (Wang et al., 2019) under both high-resource and low-resource conditions. We follow previous work (Lester et al., 2021; Vu et al., 2022) to report the performance of our model on the validation sets due to the restricted access to the test sets of SuperGLUE. We provide detailed descriptions, evaluation metrics, and statistics of the SuperGLUE benchmark in Appendix A.3.

### 4.2 Settings

We use the encoder-decoder T5 model (Raffel et al., 2020) as the backbone for our experiments. We employ three variants of the T5 model: T5-Small, T5-Base, and T5-Large, which comprise 60M, 220M, and 770M parameters respectively. We implement our approach with the HuggingFace Transformers library (Wolf et al., 2020). Each dataset within the SuperGLUE benchmark is converted into a text-to-text format for compatibility with the T5 model. The model is trained for 100 epochs with an initial learning rate of 0.3, and AdamW (Loshchilov and Hutter, 2019) is employed as the optimizer. The length of the soft prompt $c$ is fixed at 100. The embedding dimension $e$ is configured as per the model variant, being set to 512, 768, and 1024 for T5-Small, T5-Base, and T5-Large respectively. Importantly, we set the bottleneck size $b$ as 10 to achieve a balance of performance and parameter efficiency. This bottleneck plays an instrumental role in inducing low-rank constraints on the embedded soft prompt matrix $P_{emb}$. We initialize the matrices $A$ and $B$ of our prompt parameters with Gaussian distribution.

### 4.3 Baselines

To maintain consistency in evaluation metrics and preprocessing procedures across all datasets, we reproduce most of the scores to ensure a fair comparison. More details about implementation can be found in Appendix A.4.

**Fine-tuning** We compare DPT with the conventional fine-tuning approach of the T5 model (Raffel et al., 2020), where separate copies of the model must be tuned and stored for different datasets. Though fine-tuning is not a parameter-efficient ap-

| Model | # Trainable Params. | WSC Acc. | WiC Acc. | BoolQ Acc. | CB Acc. | COPA Acc. | MultiRC F1 | ReCoRD F1 | RTE Acc. | Avg. |
|---|---|---|---|---|---|---|---|---|---|---|
| | | | | | T5-Small | | | | | |
| *Fine-Tuning* | 60M | 67.94 | 68.18 | 77.06 | 89.28 | 59.00 | 66.98 | 55.64 | 72.56 | 69.58 |
| Prompt Tuning (Lester et al., 2021) | 51K | 63.14 | 59.29 | 66.78 | 74.99 | **58.33** | **64.89** | 52.75 | 59.92 | 62.51 |
| Residual PT (Razdaibiedina et al., 2023) | 462K | 63.14 | 60.96 | **73.35** | 72.02 | 56.66 | 65.12 | 53.08 | 67.02 | 63.91 |
| Ours | 6K | **63.78** | **64.31** | 71.74 | **79.16** | 58.00 | **64.89** | **53.27** | **70.51** | **65.70** |
| | | | | | T5-Base | | | | | |
| *Fine-Tuning** | 220M | 81.70 | 69.30 | 82.30 | 91.70 | 60.00 | 76.90 | 80.90 | 84.50 | 78.41 |
| Prompt Tuning (Lester et al., 2021) | 77K | 64.74 | 59.97 | 62.18 | 70.23 | 56.33 | **72.69** | 71.84 | 56.43 | 64.30 |
| Residual PT (Razdaibiedina et al., 2023) | 693K | **67.94** | 63.31 | **80.00** | 77.37 | 56.66 | 72.11 | 72.21 | 81.70 | 71.41 |
| Ours | 9K | 67.30 | **68.49** | 78.64 | **78.56** | 56.66 | 71.22 | **72.53** | **81.94** | **71.88** |
| | | | | | T5-Large | | | | | |
| *Fine-Tuning** | 770M | 88.50 | 73.50 | 88.30 | 94.30 | 87.00 | 85.40 | 89.20 | 90.60 | 87.10 |
| Prompt Tuning (Lester et al., 2021) | 102K | 76.52 | 70.00 | 84.09 | 74.40 | 62.00 | 76.18 | **84.51** | 88.95 | 77.08 |
| Residual PT (Razdaibiedina et al., 2023) | 925K | 70.50 | **72.25** | **85.04** | 73.21 | **62.66** | 76.46 | 84.36 | 88.92 | 76.67 |
| Ours | 11K | **79.16** | 71.99 | 84.76 | **88.09** | 62.33 | **76.72** | 84.46 | **90.25** | **79.72** |

Table 1: Results on the SuperGLUE validation set. All scores are the average over 3 runs with different random seeds. The last column (Avg.) indicates the average score across all datasets in the SuperGLUE benchmark. The models with * symbol denote that the results are retrieved from Aribandi et al. (2022), and the rest are reproduced by following the original implementations. Standard deviation of three runs is in Appendix A.5.

proach, it is usually treated as an upper bound of performance.

**Prompt Tuning** Prompt tuning (PT) (Lester et al., 2021) prepends a soft prompt to the input embedding. It adapts to downstream tasks by exclusively updating the parameters of the soft prompt, keeping the language model frozen.

**Residual Prompt Tuning** Residual prompt tuning (Residual PT) (Razdaibiedina et al., 2023) is a recently proposed variant of PT. It enhances the original prompt embeddings through an additional layer of shallow network instead of passing them directly into the frozen transformer layer. A residual connection is also employed to boost the performance and convergence rate.

Our proposed model is not directly comparable with the multi-stage XPrompt (Ma et al., 2022), which iteratively updates the soft prompt embedding. Adapter-based methods (Houlsby et al., 2019; Rücklé et al., 2021; Guo et al., 2021) are also not included for comparison in this paper as they require efforts to modify the transformer layers. Furthermore, they often include a considerably larger number of parameters than prompt tuning approaches as mentioned in Razdaibiedina et al. (2023). More detailed comparisons can be found in Section 7.

## 5 Results

### 5.1 Main Results

Table 1 presents a comparison of our DPT with the vanilla prompt tuning (Lester et al., 2021) and

residual prompt tuning (Razdaibiedina et al., 2023) across 8 datasets from the SuperGLUE benchmark. Besides the model performance, we also include the comparison of the total number of trainable parameters. Remarkably, our model consistently outperforms the aforementioned approaches across T5 models of different sizes in terms of the average scores over the 8 datasets. Specifically, our model outperforms the vanilla prompt tuning by 3.19, 7.58, and 2.64 points in terms of the average score (Avg.) with T5-Small, T5-Base, and T5-Large models, respectively. In most cases, our model surpasses the vanilla prompt tuning approach by a large margin. Moreover, DPT is highly efficient in terms of parameter usage, requiring approximately one-ninth of the number of trainable parameters compared to vanilla prompt tuning (i.e., 11.2K vs. 102K for T5-Large). When compared with the recently proposed residual prompt tuning, our model also shows better or comparable performance. Crucially, DPT achieves this while requiring significantly fewer trainable parameters. Specifically, residual prompt tuning consumes nearly 84 times more parameters than our method (i.e., 11.2K vs. 925K for T5-Large). Note that we set the length of the soft prompt for residual prompt tuning to 100 which empirically yields better performance [3]. We also provide the experimental results of residual prompt tuning when setting the length of the soft prompt to 10 in Appendix A.6. The experimental

---

[3]There are 100-token and 10-token residual prompt tuning variants in the original paper (Razdaibiedina et al., 2023).

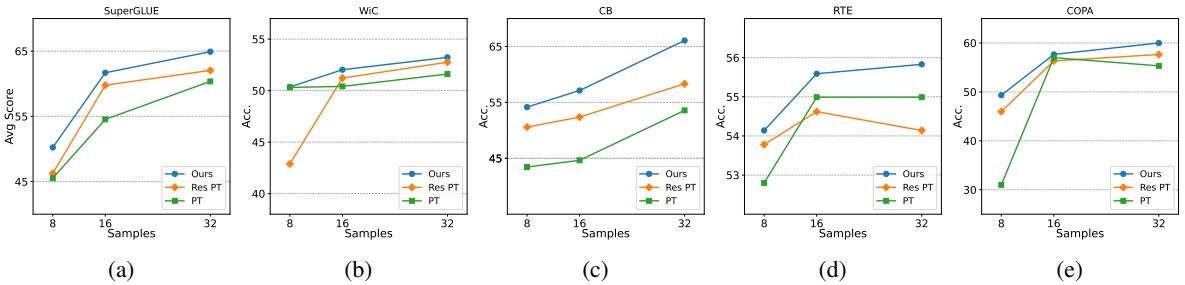

Figure 4: Comparison of DPT with vanilla prompt tuning (PT) and Residual PT (Res PT) in the few-shot setting.

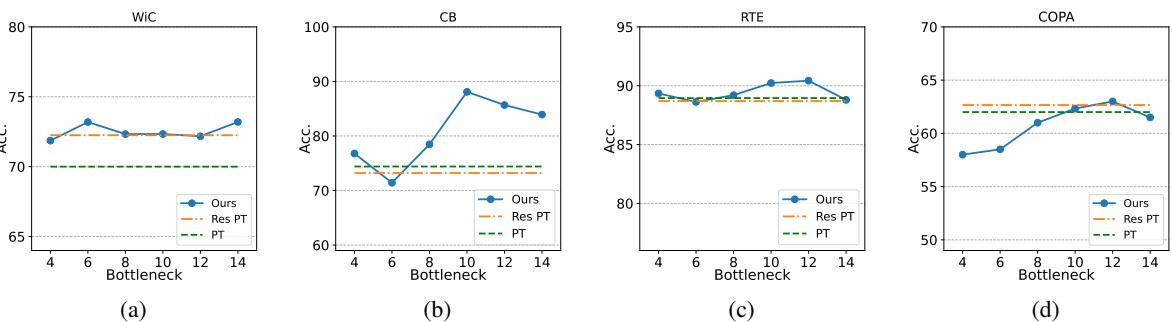

Figure 5: Comparisons of DPT with PT and Res PT with different sizes of the bottleneck $b$. Each point represents the average of three runs with different random seeds.

results demonstrate the effectiveness of our proposed low-rank soft prompt in both performance and parameter efficiency.

## 5.2 Few-shot Performance

We further evaluate our method in a few-shot setting on all datasets in the SuperGLUE benchmark. Specifically, we sample total of 8, 16, and 32 training instances without balancing the label distribution to mimic a realistic scenario. To ensure a fair comparison, the sampled data for our method, vanilla prompt tuning, and residual prompt tuning are kept identical for each run. We employ the T5-Large model as the backbone for this evaluation as it achieves the strongest performance in the vanilla prompt tuning. The results presented are averaged across three separate runs with different random seeds. We compare our approach against the previous methods in terms of the average score over all datasets in the SuperGLUE benchmark, and the results are shown in Figure 4a. We observe that our method consistently outperforms both vanilla prompt tuning and residual prompt tuning in our experiments. Additionally, We also include experimental results on specific datasets from Super-GLUE in Figure 4b, 4c, 4d, and 4e. These results further demonstrate the effectiveness of our method over the baselines in the few-shot setting.

## 6 Analysis

### 6.1 Sensitivity of Bottleneck Size

As mentioned in Section 3.1, the rank of our soft prompt can be adjusted through the bottleneck $b$. In this section, we investigate how the size of the bottleneck impacts the performance of DPT. We conduct experiments using T5-Large on the WiC, CB, RTE, and COPA datasets, with bottleneck $b \in \{4, 6, 8, 10, 12, 14\}$. The results are shown in Figure 5. We can see that though our approach experiences performance fluctuations with different sizes of the bottleneck $b$, DPT outperforms the vanilla prompt tuning and residual prompt tuning in quite a few cases. Intuitively, the size of the bottleneck plays a critical role in the expressiveness of our model. We observe a performance drop when the size of the bottleneck is getting smaller. We find similar behavior when T5-Base is used, and the results can be found in Appendix A.7.

### 6.2 Effect of Prompt Length

To investigate the effect of prompt length on the performance of our proposed method, we conduct an analysis on four of the SuperGLUE tasks[4], and the results are shown in Figure 6. The prompt

---

[4]These four datasets are randomly selected for analysis.

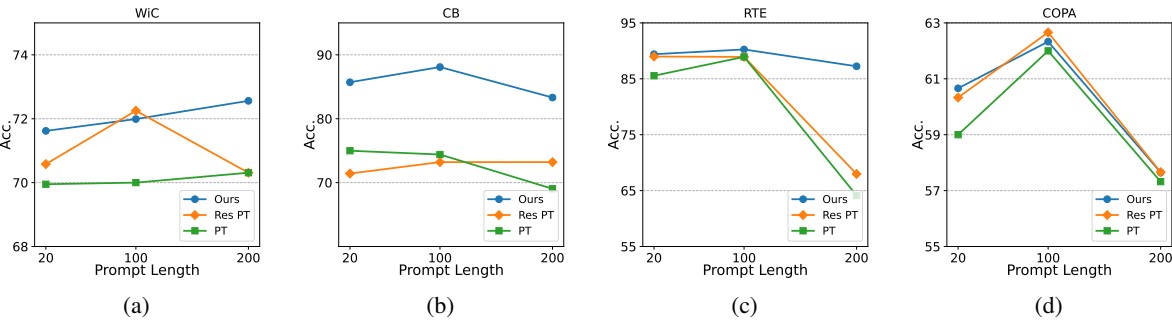

Figure 6: Comparisons of our method with PT and Res PT with different prompt lengths $c \in \{20, 100, 200\}$. Each point is the average of three runs with different random seeds.

length $c$ is set as 20, 100, and 200. We use the T5-Large model as the backbone, while the bottleneck variable $b$ is held constant at 10. As mentioned in Section 3, the number of trainable parameters in the vanilla PT is $ec$, whereas for our DPT method, it is $eb + bc$. The benefit of our approach in terms of parameter-efficiency is more pronounced as the prompt length increases[5]. We observe that our DPT consistently outperforms the vanilla prompt tuning approach across various prompt lengths. More-over, DPT is generally found to be comparable or superior to residual prompt tuning. Corroborat-ing the observations made by Lester et al. (2021), a long soft prompt does not necessarily improve the model performance. While a prompt length of 100 has been empirically validated as an ef-fective hyperparameter in previous work (Lester et al., 2021; Razdaibiedina et al., 2023), our analy-sis reveals that the performance gains can still be achieved with a prompt length of 200, as illustrated in Figure 6a. Furthermore, our DPT is less sen-sitive to the change of prompt length on CB and RTE, as shown in Figure 6b and Figure 6c. We observe similar behavior when switching to T5-Base as the backbone, and details are included in Appendix A.8.

### 6.3 Constrained Bottleneck under Short Prompt

A critical aspect we sought to address with DPT is the potential to sustain both parameter efficiency and competitive performance, even with a short prompt. When the prompt length $c$ is reduced to values such as 6 or 10, DPT may not exhibit param-eter efficiency if the bottleneck $b$ remains to be 10. To explore the possibility of achieving parameter

---

[5]When $b$ and $e$ are fixed, the ratio $\frac{eb+bc}{ec}$ decreases as the prompt length $c$ increases. This implies that our method ex-hibits higher parameter efficiency with longer prompt lengths.

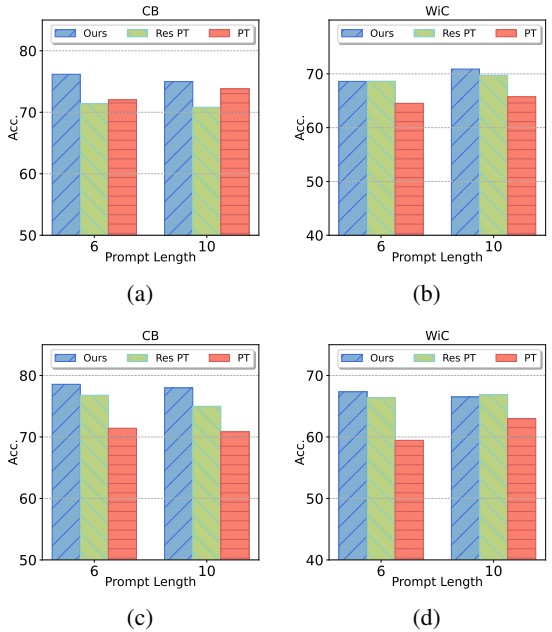

Figure 7: Further comparisons with short prompt lengths and small bottleneck. 7a and 7b are based on T5-Large while 7c and 7d are based on T5-Base.

efficiency under such conditions, we compress the bottleneck to an extremely low value of 2. The experiments conducted on the CB and WiC tasks using the T5-Large and T5-Base model are illus-trated in Figure 7. Notably, even with the bottle-neck compressed to such a small value, our method is still able to surpass baselines. This demonstrates that our method is able to effectively generalize to extremely short prompts with a very constrained bottleneck.

### 6.4 Bottleneck vs. Number of Trainable Parameters

To further study the effect of bottleneck size with-out considering parameter efficiency, we conduct experiments with the bottleneck $b$ of varying mag-

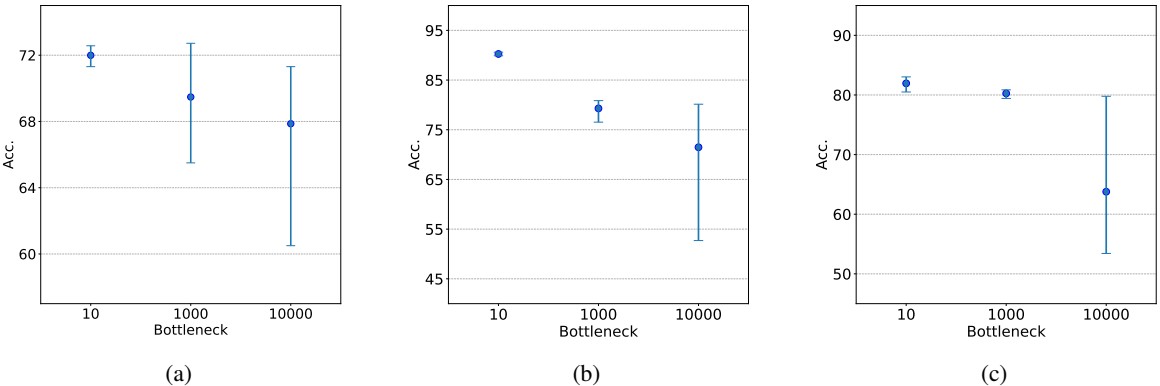

Figure 8: We enlarge the bottleneck to even 10000 to study the performance of DPT. $x$-axis is the bottleneck size. $y$-axis is the performance. 8a and 8b are from T5-Large while 8c is from T5-Base.

nitudes, specifically $b \in \{10, 1000, 10000\}$, while maintaining a fixed prompt length of 100 tokens. The experimental results are on the RTE dataset with the T5-Large and T5-Base model, and we record the average, minimal, and maximum scores across three runs. Figure 8 shows the comparison, and it is noteworthy that the number of trainable parameters involved are 11.2K, 1.1M, and 11.2M respectively for the varying bottleneck sizes under T5-Large. When configuring the bottleneck $b$ to a large value, the soft prompt ceases to be a low-rank matrix and the number of trainable parameters is more than that of a vanilla prompt tuning approach. More importantly, we observe that increasing the number of trainable parameters does not necessarily lead to an enhancement in performance. Furthermore, overparameterization could deteriorate the stability of the performance.

## 7 Related Work

Parameter-efficient fine-tuning (PEFT) methods (He et al., 2022a; Ben Zaken et al., 2022; He et al., 2022b; Mao et al., 2022; He et al., 2022c) have emerged as a popular approach to fine-tune language model which can largely reduce the number of trainable parameters while maintaining competitive performance. There are two primary paradigms for current PEFT methods: adapter-based and prompt-based approaches.

**Adapter-based Methods** The concept of the adapter was originally proposed by Houlsby et al. (2019). They inserted a down projection layer followed by an up projection layer in each sub-module of the transformer sequentially. When adapting a model to downstream tasks, only the parameters

of the adapter were updated (Pfeiffer et al., 2021; Sung et al., 2022; Chen et al., 2023; Zeng et al., 2023). To enhance this, Karimi Mahabadi et al. (2021) incorporated hypernetworks, which generate weights for the main network, thus enabling shared information capture across tasks. Following that, Hu et al. (2022) proposed LoRA to approximate the update of the neural weight. LoRA was based on the assumption that the change in weights during model adaptation has a low "intrinsic rank". Different from the adapter, LoRA did not introduce additional latency. Apart from being a PEFT method, the adapter has also been employed in broader applications in natural language processing (Pfeiffer et al., 2021).

Our work diverges from LoRA (Hu et al., 2022) and the broader adapter framework. Rather than learning updates to the parameters, our approach directly learns the parameters, under the hypothesis that the soft prompt inherently exhibits a low "intrinsic rank". Furthermore, LoRA focuses on Query-specific and Value-specific parameters within the transformer architecture, whereas our approach adheres to the prompt tuning paradigm, where trainable parameters are exclusively inserted in the embedding layer. Despite these fundamental differences, it is noteworthy that both our approach and LoRA share an underlying principle of leveraging low-rank structures.

**Prompt-based Methods** Prompt-based methods can be categorized into hard prompt and soft prompt approaches. Hard prompts involve adding a fixed sequence of discrete tokens for the model to condition on for generation (Jiang et al., 2020; Shin et al., 2020; Zhao et al., 2021; Liu et al., 2023).

However, they are sensitive to slight changes and require complex designs, like verbalizer selection. To address this, trainable soft prompt methods were introduced. Li and Liang (2021) introduced prefix tuning, which adds virtual tokens in each layer of the encoder stack. As a simplification of prefix tuning, Prompt Tuning (Lester et al., 2021) only adds a soft prompt to the embedding layer. Further advancements include incorporating soft prompts across all transformer layers to enhance performance (Liu et al., 2022b). Additionally, transfer learning-based methods (Wang et al., 2023; Vu et al., 2022) have been explored for better prompt initialization through pre-training. For instance, Wang et al. (2023) performed pre-training to learn a soft prompt on a collection of source tasks and subsequently employed the acquired prompt embeddings as initialization for target tasks. Transfer learning-based techniques offer the potential for better initialization and more effective prompt tuning. In contrast to prior work, this paper focuses on developing a more efficient parameterization of the soft prompt.

## 8 Conclusion

In this work, we uncover the low "intrinsic rank" behavior inherent in the soft prompt through empirical examination. Motivated by these findings, we introduce *Decomposed Prompt Tuning* (DPT), a novel approach that reparameterizes the soft prompt using two compact matrices. By adjusting the bottleneck, DPT enables the effective manipulation of the soft prompt matrix, ensuring it maintains a low rank. Notably, DPT attains strong performance across the SuperGLUE benchmark in high-resource and low-resource scenarios while substantially reducing the number of trainable parameters.

## Limitations

Despite our proposed approach being simple but effective, our method still possesses a few limitations. We mainly evaluate our model on the natural understanding tasks, i.e., SuperGLUE. We do not evaluate our proposed method on the natural language generation tasks. Furthermore, our method also inherits a few drawbacks from the vanilla prompt tuning, such as slow convergence. These limitations serve as future directions for further improvement. Another limitation is that we only evaluate our method with the encoder-decoder backbone

models. We leave the explorations with encoder-only models and other large-scale pre-trained models for future work.

## Acknowledgements

We would like to thank the anonymous reviewers, our meta-reviewer, and senior area chairs for their constructive comments and support on this work. This research/project is supported by Ministry of Education, Singapore, under its Tier 3 Programme (The Award No.: MOET320200004), and Ministry of Education, Singapore, under its Academic Research Fund (AcRF) Tier 2 Programme (MOE AcRF Tier 2 Award No: MOE-T2EP20122-0011). Any opinions, findings and conclusions or recommendations expressed in this material are those of the authors and do not reflect the views of the Ministry of Education, Singapore.

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

# A Appendix

## A.1 Rank Explanation

**Theorem A.1.** *Let A and B be matrices such that the product AB is well defined. Then*

$$rank(AB) \leqslant \min(rank(A), rank(B))$$

*Proof.* Each column of $AB$ is a combination of the columns of $A$, which implies that $\mathcal{R}(AB) \subseteq \mathcal{R}(A)$. Hence, $\dim(\mathcal{R}(AB)) \leq \dim(\mathcal{R}(A))$, or equivalently, $rank(AB) \leq rank(A)$ Each row of $AB$ is a combination of the rows of $B \rightarrow$ rowspace $(AB) \subseteq$ rowspace $(B)$, but the dimension of rowspace = dimension of column space = rank, so that $rank(AB) \leq rank(B)$. □

According to theorem A.1, if $P = U\Sigma V$, it can be easily obtained that:

$$rank(P) \leqslant \min(rank(U), rank(\Sigma), rank(V))$$

$rank(\Sigma)$ is an upper bound of $rank(P)$, because $U$ and $V$ are always full rank during training process. Thus, we can use $rank(\Sigma)$ as an approximation for $rank(P)$.

## A.2 Record of Rank Change

We conduct additional experiments with different models and datasets to probe the rank change of soft prompt. We find consistent observations that the soft prompt tends to exhibit a low "intrinsic rank" behavior. As we can see from Figure 9, the number of positive entries is decreasing while the number of negative entries is increasing. In the second row, we also record the rank change of the soft prompt. Specifically, we calculate the rank of the matrix obtained by multiplying $U$, $\Sigma$, and $V$.

## A.3 Dataset Details

We record the statistical details of the SuperGLUE benchmark and the metrics we use, and Table 2 shows the statistics. Note that the original CB dataset has 554 training samples, we follow (Raffel et al., 2020) to keep only 259 samples with a "True" label. We transform the task to a text-to-text format. Specifically, we highlight the ambiguous pronoun in the input text and ask the model to predict the noun that it refers to (Raffel et al., 2020).

## A.4 Baseline Implementation Details

The length of the soft prompt is set as 100 for vanilla prompt tuning, which is an optimal setting according to the paper (Lester et al., 2021). The soft prompt can be initialized by Gaussian distribution or copying from the vocabulary embedding. Here, we adopt the second initialization strategy because of its superiority.

There are 10-token and 100-token residual prompt tuning variants (Razdaibiedina et al., 2023). We implement the 100-token one because it has

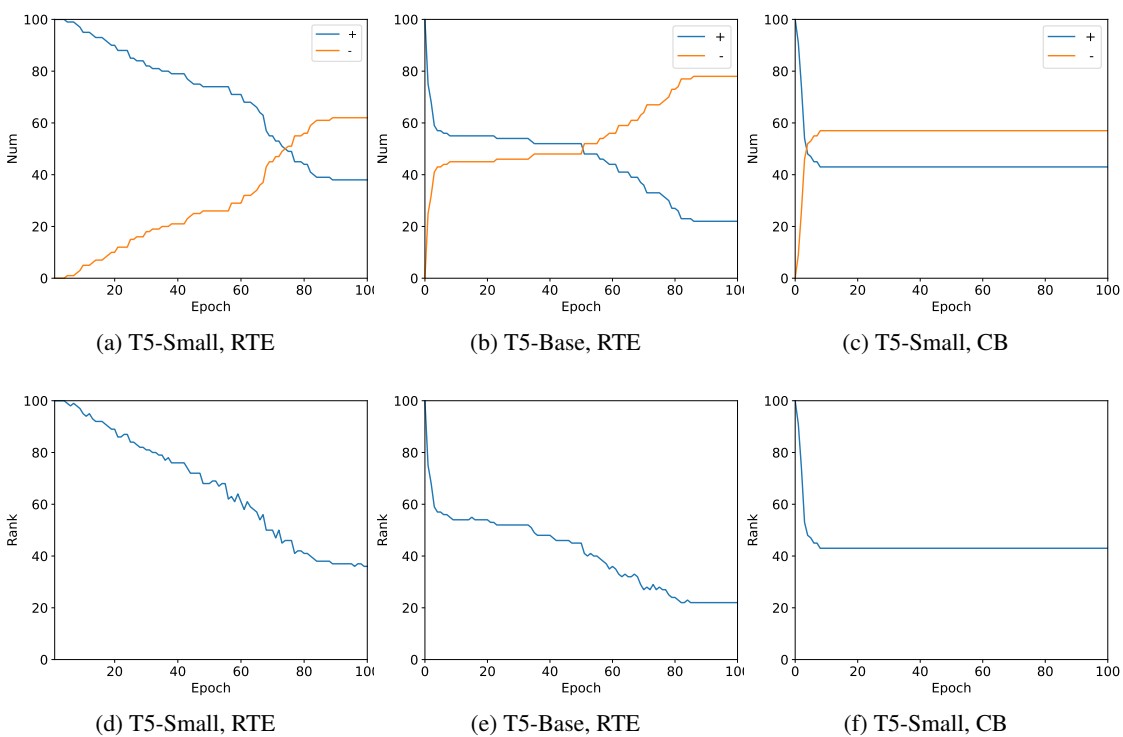

Figure 9: The first row is record of positive and negative diagonal entries. The second row is the rank record.

| Dataset | Train | Dev | Task | Metric |
|---------|-------|-----|------|--------|
| BoolQ | 9,427 | 3,270 | QA | Acc. |
| CB | 250 | 56 | NLI | Acc. |
| COPA | 400 | 100 | QA | Acc. |
| MultiRC | 27,243 | 4,848 | QA | F1 |
| ReCoRD | 100,730 | 10,000 | QA | F1 |
| WiC | 5,428 | 638 | WSD | Acc. |
| WSC | 259 | 104 | Coref. | Acc. |
| RTE | 2,490 | 277 | NLI | Acc. |

Table 2: The details of 8 SuperGLUE tasks used in our experiments.

better performance. We also follow the paper to set the bottleneck as 400, above which leads to no improvement according to the paper. We follow them to adopt ReLU as activation function and perform layer normalization (Ba et al., 2016).

### A.5 Standard Deviation of Scores

We report the the standard deviation of three runs for our method, prompt tuning and residual prompt tuning. The results are shown in Table 3.

### A.6 Residual PT of 10 tokens

In most parts of this paper, we focus on soft prompt of 100 tokens. (Razdaibiedina et al., 2023) present 10-token and 100-token variant of residual prompt tuning in their main results. We reproduce the 100-token residual prompt tuning in the main result section. Here, we also reproduce the 10-token one in Table 4 . The results are consistent with the original paper that the performance of 10-token version is inferior to that of 100-token version.

### A.7 Various Bottleneck Size

We try different bottleneck $b \in \{4, 6, 8, 10, 12, 14\}$ and fix prompt length as 100. The result of T5-Base is similar to that of T5-Large. DPT can surpass prompt tuning in most cases, which demonstrates the robustness of DPT. In Figure 10, we show the result on T5-Base.

### A.8 Various Prompt Length

We try prompt length 20, 100 and 200 and fix the bottleneck as 10. The result of T5-Base is similar to that of T5-Large. The performance of DPT is consistently better than PT under various prompt lengths as shown in Figure 11.

| Model | WSC | WiC | BoolQ | CB | COPA | MultiRC | ReCoRD | RTE |
|---|---|---|---|---|---|---|---|---|
| | | | | T5-Small | | | | |
| PT | 1.10 | 2.82 | 1.28 | 3.57 | 1.15 | 0.90 | 0.30 | 3.12 |
| Res PT | 1.10 | 5.7 | 1.64 | 4.12 | 0.57 | 0.51 | 0.22 | 3.68 |
| Ours | 0.55 | 0.65 | 0.85 | 4.12 | 4.35 | 0.90 | 0.15 | 1.98 |
| | | | | T5-Base | | | | |
| PT | 1.10 | 6.30 | 0.01 | 1.02 | 1.15 | 0.24 | 0.09 | 0.20 |
| Res PT | 0.55 | 3.68 | 10.11 | 10.46 | 2.08 | 0.20 | 0.13 | 1.50 |
| Ours | 0.96 | 1.33 | 0.79 | 6.43 | 2.08 | 0.46 | 0.10 | 1.30 |
| | | | | T5-Large | | | | |
| PT | 1.92 | 1.04 | 0.82 | 1.03 | 0.00 | 0.18 | 0.09 | 0.20 |
| Res PT | 3.08 | 0.56 | 0.13 | 1.79 | 1.52 | 0.04 | 0.17 | 0.75 |
| Ours | 1.46 | 0.63 | 0.27 | 2.06 | 2.51 | 0.39 | 0.15 | 0.36 |

Table 3: We report standard deviation (%) of three runs for PT, Res PT and our method.

| Model | # Trainable Params. | WSC Acc. | WiC Acc. | BoolQ Acc. | CB Acc. | COPA Acc. | MultiRC F1 | ReCoRD F1 | RTE Acc. | Avg. |
|---|---|---|---|---|---|---|---|---|---|---|
| | | | | | T5-Small | | | | | |
| Residual PT | 416K | 64.73 | 58.09 | 69.70 | 73.21 | 58.66 | 63.21 | 52.82 | 62.21 | 62.83 |
| | | | | | T5-Base | | | | | |
| Residual PT | 624K | 68.26 | 66.87 | 79.11 | 74.99 | 57.66 | 71.20 | 72.41 | 68.58 | 69.88 |
| | | | | | T5-Large | | | | | |
| Residual PT | 832K | 69.86 | 69.74 | 84.18 | 70.82 | 59.33 | 74.77 | 84.31 | 88.20 | 75.15 |

Table 4: Results on the SuperGLUE validation set. All scores are the average over 3 runs with different random seeds. The last column (Avg.) indicates the average score across all datasets in the SuperGLUE benchmark.

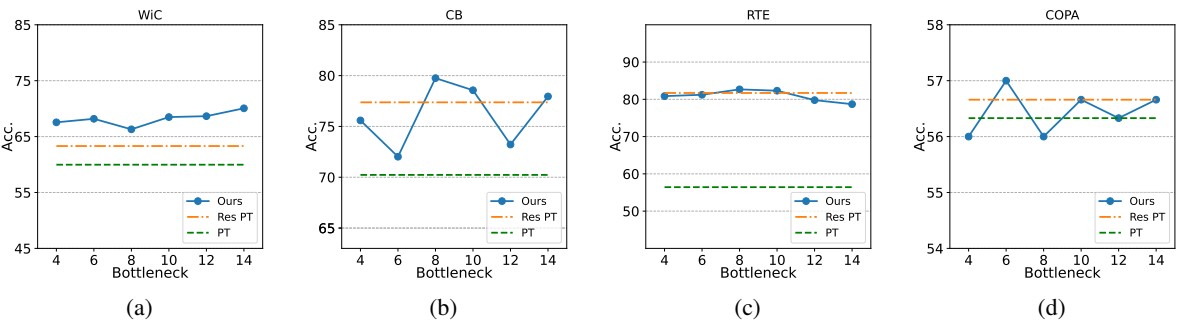

Figure 10: Comparisons of DPT with PT and Res PT with different sizes of the bottleneck $b$. Each point represents the average of three runs with different random seeds.

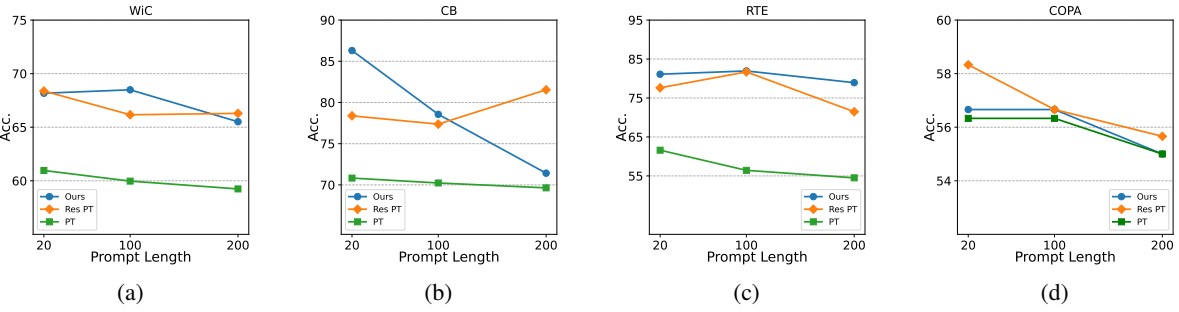

Figure 11: The comparison between our method and prompt tuning of different length in $\{20, 100, 200\}$. Each point in the figure is average of three runs with random seeds. The backbone is T5-Base.