# OpenReview forum: "Decomposed Prompt Tuning via Low-Rank Reparameterization"
_EMNLP/2023/Conference — EMNLP 2023 Findings_

### Official Review · Reviewer_4fcj · 2023-08-03

**Soundness:** 3

**Excitement:**

3: Ambivalent: It has merits (e.g., it reports state-of-the-art results, the idea is nice), but there are key weaknesses (e.g., it describes incremental work), and it can significantly benefit from another round of revision. However, I won't object to accepting it if my co-reviewers champion it.

**Missing References:**

The authors have appropriately and comprehensively included relevant references in the bibliography.

**Paper Topic And Main Contributions:**

The paper introduces an innovative method that employs low-rank matrices to formulate the soft prompt, which can enhance the efficiency of prompt tuning. This approach not only improves the performance but also reduces the training parameters.

**Questions For The Authors:**

Question A: Could you provide further empirical or theoretical analyses to explain why the decomposed prompt works effectively?
Question B: The optimization of prompt tuning is typically challenging. Could you elaborate on how your approach addresses this issue?

**Reasons To Accept:**

The paper's strength lies in its innovative approach to constructing the soft prompt through low-rank reparameterization. This concept contributes to a deeper understanding of parameter-efficient fine-tuning in prompt tuning. Additionally, the paper provides substantial experimental evidence to support the efficacy of the proposed method.

**Reasons To Reject:**

While the paper presents an intriguing method and boasts compelling experimental results, the authors fail to clearly explain the relationship between the rank and the soft prompt. The experimental results and analysis do not appear sufficiently robust to draw firm conclusions.

**Reproducibility:**

4: Could mostly reproduce the results, but there may be some variation because of sample variance or minor variations in their interpretation of the protocol or method.

**Reviewer Confidence:**

4: Quite sure. I tried to check the important points carefully. It's unlikely, though conceivable, that I missed something that should affect my ratings.

**Typos Grammar Style And Presentation Improvements:**

The paper is well-written with no discernible typographical or grammatical errors. The style and presentation of the research are clear and easy to follow.

---

> ### Author Rebuttal · Authors · 2023-08-29
>
> Reviewer 4fcj:
>
> Thank you for your thoughtful review. We appreciate your recognition of our contributions and will address your concerns and questions.
>
> Question A: Could you provide further empirical or theoretical analyses to explain why the decomposed prompt works effectively?
>
> In Section 2, we conduct an empirical analysis to investigate the rank of soft prompt. Through re-designing the soft prompt, our experiments facilitate a probe into its rank. We further provide a mathematical explanation for using the rank of Relu(Σ) as an approximation for the rank of P_emb.
>
> Again, here is a short explanation. First, we reparameterize soft prompt P_emb by the form: P_emb = U∙Relu(Σ)∙V. U and V are trainable parameters that are randomly initialized. We initialize the trainable diagonal entries of Σ with positive values while fixing other entries as 0.
> Through such a design, the soft prompt can change its rank more freely.
>
> During the training process, U and V are always full rank, which means Rank(Relu(Σ))  is the upper bound of Rank(P_emb).  Then, we use Rank(Relu(Σ)) as an approximation of Rank(P_emb). We provide proof for this in Appendix A.1. Intuitively if soft prompt exhibits a low “intrinsic rank”, it will update more and more diagonal entries of Σ to be negative (After Relu activation, it will be zero).
> We plot the number of positive and negative diagonal entries of  Σ  with the training proceeding in Figure 3. Furthermore, we also directly compute the rank of P_emb after every epoch, and its pattern is identical to the curve of the number of positive diagonal entries in Figure 3. We provide more figures in Appendix A.2 of the paper.
>
> Overall, the rank of P_emb and the number of positive diagonal entries of Σ are decreasing with training proceeding. We think those theoretical analyses and empirical records can serve as evidence to support our assumption that soft prompt exhibits a low “intrinsic rank”, which motivates us to reparameterize the soft prompt with two low-rank matrices explicitly. This also helps verify the effectiveness of our method.
>
> Question B: The optimization of prompt tuning is typically challenging. Could you elaborate on how your approach addresses this issue?
>
> Answer: The main contribution of our paper is that we propose a parameter-efficient training approach with a much smaller number of trainable parameters (i.e., 1/9 of prompt tuning, 1/84 of residual prompt tuning for T5-large), while achieving better performance. The discussion of optimization issues is not the main focus of this paper. However, we do try different learning rate decay strategies like warmup and annealing for our experiments, but they are not as effective as expected. We find training for longer time like 100 epochs as prompt tuning [1] did is a good way to obtain stable performance.
>
> [1] The Power of Scale for Parameter-Efficient Prompt Tuning
>
> Reason to reject: While the paper presents an intriguing method and boasts compelling experimental results, the authors fail to clearly explain the relationship between the rank and the soft prompt. The experimental results and analysis do not appear sufficiently robust to draw firm conclusions.
>
> Answer: As we have restated in the reply to answer A, we empirically show that soft prompt has a low “intrinsic rank”, which motivates us to reparameterize the soft prompt with low-rank matrices. We find that a bottleneck around 10 helps us achieve a balance between parameter efficiency and performance (in the main experiment, we use bottleneck 10).

---

### Official Review · Reviewer_JCRp · 2023-08-05

**Soundness:** 4

**Excitement:**

3: Ambivalent: It has merits (e.g., it reports state-of-the-art results, the idea is nice), but there are key weaknesses (e.g., it describes incremental work), and it can significantly benefit from another round of revision. However, I won't object to accepting it if my co-reviewers champion it.

**Paper Topic And Main Contributions:**

This paper focus on improving prompt tuning approaches for large language models, particularly in the context of high efficiency and effectiveness, by decomposing the soft prompt into low-rank matrices, which reduces the number of trainable parameters.

**Reasons To Accept:**

The experiment results are promising according to the paper, by finetuning far less trainable parameters and achieving better performance. Meanwhile, the similar performance can not be gained through simply reducing the soft prompt length. The analysis of the proposed method is sufficient.

**Reasons To Reject:**

Although the decrease of the number of tuning parameters seems promising, I have noticed that the setup of the experiments needs 100 epochs, while in Residual Prompt tuning [2], the training epoch is 20. This is in accordance with the conclusion from [1], that although soft prompt tuning is efficient in memory, it is not efficient in training time, usually suffering from longer convergence. I think in practice, the trainable parameters are whether 10K or 100K, make no significant difference since the memory is not bottleneck, but increasing the training time to 5x may be unaffordable.



[1]: Lialin V, Deshpande V, Rumshisky A. Scaling down to scale up: A guide to parameter-efficient fine-tuning[J]. arXiv preprint arXiv:2303.15647, 2023.

**Reproducibility:**

4: Could mostly reproduce the results, but there may be some variation because of sample variance or minor variations in their interpretation of the protocol or method.

**Reviewer Confidence:**

4: Quite sure. I tried to check the important points carefully. It's unlikely, though conceivable, that I missed something that should affect my ratings.

---

> ### Author Rebuttal · Authors · 2023-08-29
>
> Reviewer JCRp:
>
> Thank you for your thoughtful comments. We appreciate your valuable feedback and will address the questions and concerns raised.
>
> Regarding the weakness of training efficiency:
>
> To have a fair comparison, we follow the original prompt tuning [1] to set the number of training epochs as 100 for all the reported methods in our paper.  Even though the Res PT [2] used 20 epochs for their experiments,  we find that training for 100 epochs leads to further performance improvement. We attach average scores of 3 runs of Res PT on CB, RTE, and WiC datasets with T5-large below, training for 20 epochs and 100 epochs respectively.
>
> | Epoch            | 20 | 100 |
> |-------------------|-------------|----------------------|
> | CB                | 57.13       | 73.21                |
> | RTE               | 59.80       | 88.92                |
> | WiC               | 69.37       | 72.25                |
>
> Note that our method consumes fewer trainable parameters (i.e., 1/84 of  Res PT for T5-Large ) while achieving better performance.
>
> [1] The Power of Scale for Parameter-Efficient Prompt Tuning
> [2] Residual Prompt Tuning: Improving Prompt Tuning with Residual Reparameterization

---

### Official Review · Reviewer_1oRu · 2023-08-11

**Soundness:** 4

**Excitement:**

4: Strong: This paper deepens the understanding of some phenomenon or lowers the barriers to an existing research direction.

**Paper Topic And Main Contributions:**

This paper proposes a novel approach called Decomposed Prompt Tuning via Low-Rank Reparameterization to derive soft prompts that significantly reduces the number of trainable parameters while maintaining effectiveness. The paper addresses the problem of high computational cost and memory requirements associated with prompt tuning, which is a popular technique for fine-tuning large pre-trained language models for downstream tasks. The main contributions of the paper are:

1. A novel approach to derive soft prompts that reduces the number of trainable parameters while maintaining effectiveness.
2. A low-rank reparameterization technique that helps reduce the number of trainable parameters.
3. Experimental results on the SuperGLUE benchmark in both high-resource and low-resource scenarios, demonstrating the effectiveness of the proposed method.
4. Comparison with existing methods, showing that the proposed method outperforms them in terms of both effectiveness and efficiency.

The paper falls under the category of approaches for data- and compute efficiency, and it provides a solution to the problem of high computational cost and memory requirements associated with prompt tuning. The paper's contributions are significant and relevant to the NLP community, and the proposed method has the potential to improve the efficiency and effectiveness of prompt tuning for fine-tuning large pre-trained language models for downstream tasks.

**Reasons To Accept:**

The strengths of this paper are:

1. The proposed approach is simple but effective, and it significantly reduces the number of trainable parameters while maintaining effectiveness.
2. The low-rank reparameterization technique is a novel and effective way to reduce the number of trainable parameters.
3. The experimental results on the SuperGLUE benchmark in both high-resource and low-resource scenarios demonstrate the effectiveness of the proposed method.
4. The proposed method outperforms existing methods in terms of both effectiveness and efficiency.
5. The paper is well-written, and the experimental results are presented clearly and comprehensively.

**Reasons To Reject:**

According to the conference policies, weaknesses should not include factors such as the paper's length, the number of citations, or the authors' affiliations. Based on this, the potential weaknesses of this paper are:

1. The proposed method is evaluated only on the SuperGLUE benchmark, and it is unclear how well it would perform on other datasets or tasks.
2. The paper does not provide a detailed analysis of the computational cost and memory requirements of the proposed method, which could limit its practical applicability.
3. The paper does not provide a thorough comparison with other related methods, which could limit the understanding of the proposed method's strengths and weaknesses.
4. The paper does not provide a detailed analysis of the interpretability of the derived soft prompts, which could limit the understanding of the model's decision-making process.

**Reproducibility:**

4: Could mostly reproduce the results, but there may be some variation because of sample variance or minor variations in their interpretation of the protocol or method.

**Reviewer Confidence:**

3: Pretty sure, but there's a chance I missed something. Although I have a good feel for this area in general, I did not carefully check the paper's details, e.g., the math, experimental design, or novelty.

---

> ### Author Rebuttal · Authors · 2023-08-29
>
> Reviewer 1oRu:
>
> Thank you for your insightful review. We appreciate your valuable feedback and will address the concerns raised.
>
> Regarding weakness 1 about our experimental settings:
>
> We follow the previous parameter-efficient finetuning work [1] [2] [3] to evaluate the model’s performance on the SuperGLUE benchmark. Note that SuperGLUE contains eight datasets that cover natural language inference, conference resolution, question answering, and word sense disambiguation. Furthermore, to verify the generalization of our model, we also include the experimental results with different models, such as T5-small, T5-base, and T5-large. Therefore, we believe our proposed approach could generalize to more tasks.
>
> [1] The Power of Scale for Parameter-Efficient Prompt Tuning
> [2] SPoT: Better Frozen Model Adaptation through Soft Prompt Transfer
> [3] Residual Prompt Tuning: Improving Prompt Tuning with Residual Reparameterization
>
> Regarding weakness 2 about the computational cost and memory:
>
> We follow previous PEFT work to mainly focus on the comparison of number of the trainable parameters and performance. The amount of computation cost and the number of trainable parameters is generally positively correlated. Similarly, more trainable parameters also lead to more memory consumption, because the parameters and gradients are required to be stored when training. Therefore, the computational cost and memory requirement is in descending order for Res PT, PT, and our method. For instance, the stable memory requirement is 25108MB, 17086 MB, 10756 MB, and 10754 MB, for Full Finetuning, Res PT, PT, and our method respectively. The effect of pursuing the extremely limited number of trainable parameters is not conspicuous in memory reduction. Even if we decrease the number of trainable parameters to 1, it still requires about 10754 MB of memory.
>
>
> Regarding the weakness of more related work:
>
> We mainly focus on the variants of prompt tuning in this paper. We compare our method with original prompt tuning and the latest variant of prompt tuning (Res PT). As mentioned in Section 7 Related Work, we didn’t include adapter-based approaches [1] [2], which consume many more parameters, considering the up-projection and down-projection layers are required in every block of the transformer. We did not include work like [3][4] which are also prompt variants. This is because they emphasize the transfer ability of soft prompt. The advantage of our method is that it achieves promising performance with a small amount of trainable parameters.
>
> [1] LoRA: Low-Rank Adaptation of Large Language Models
> [2] Parameter-Efficient Transfer Learning for NLP
> [3] Multitask Prompt Tuning Enables Parameter-Efficient Transfer Learning
> [4] ATTEMPT: Parameter-Efficient Multi-task Tuning via Attentional Mixtures of Soft Prompts
>
> Regarding the weakness of interpretability of the derived soft prompts:
>
> We consider the soft prompt as a continuous version of the hard prompt. Both are trying to elicit knowledge from language models for downstream tasks. However, the soft prompt is less interpretable than the hard prompt since the embedding space is continuous and much more complex.

---

### Meta-Review · Area_Chair_zFS9 · 2023-09-18

**Recommendation:** 4

**Metareview:**

The paper presents a novel method, "Decomposed Prompt Tuning via Low-Rank Reparameterization," aiming to enhance prompt tuning efficiency for large pre-trained language models. By decomposing the soft prompt into low-rank matrices, the proposed approach reduces trainable parameters in the prompt matrix. The approach is experimentally validated using the SuperGLUE benchmark across various-sized T5 models.

Generally, the reviewers responded positively to the parameter reduction achieved by the method while maintaining similar or better performance. Additionally, the use of low-rank reparameterization seems well motivated in this domain.

There were concerns raised about the lack of comparison to other adapter-based approaches like LoRA. In the rebuttal, the authors argue that these approaches are less parameter efficient than prompt tuning. However, this argument feels a bit weak to me for multiple reasons. First, the authors compare against Fine-Tuning, which requires training the entire model (many more parameters than a LoRA). Second, and more importantly in my mind, the motivation to compare different parameter efficient fine-tuning methods is to see the trade-off across the additional parameters, computational cost, and performance of each method.

For instance, while LoRAs may require more parameters, all of these methods likely contribute to a fraction of the storage and computation requirements of the larger LLM they are augmenting. This makes the reduction in memory (even a 10x reduction) harder to quantify without additional results to better contextualize it. The authors may include some additional empirical observations (and comparison to other existing methods) along these lines to better show how this work is positioned in this space.

---

### Decision · Program_Chairs · 2023-10-07

**Decision:**

Accept-Findings

**Comment:**

The paper presents a novel method, "Decomposed Prompt Tuning via Low-Rank Reparameterization," aiming to enhance prompt tuning efficiency for large pre-trained language models. By decomposing the soft prompt into low-rank matrices, the proposed approach reduces trainable parameters in the prompt matrix. The approach is experimentally validated using the SuperGLUE benchmark across various-sized T5 models.

Generally, the reviewers responded positively to the parameter reduction achieved by the method while maintaining similar or better performance. Additionally, the use of low-rank reparameterization seems well motivated in this domain.

There were concerns raised about the lack of comparison to other adapter-based approaches like LoRA. In the rebuttal, the authors argue that these approaches are less parameter efficient than prompt tuning. However, this argument feels a bit weak to me for multiple reasons. First, the authors compare against Fine-Tuning, which requires training the entire model (many more parameters than a LoRA). Second, and more importantly in my mind, the motivation to compare different parameter efficient fine-tuning methods is to see the trade-off across the additional parameters, computational cost, and performance of each method.

For instance, while LoRAs may require more parameters, all of these methods likely contribute to a fraction of the storage and computation requirements of the larger LLM they are augmenting. This makes the reduction in memory (even a 10x reduction) harder to quantify without additional results to better contextualize it. The authors may include some additional empirical observations (and comparison to other existing methods) along these lines to better show how this work is positioned in this space.